# Investigating the Association between Farmers' Organizational Participation and Types of Agricultural Product Certifications: Empirical Evidence from a National Farm Households Survey in Taiwan

**Mei-Yin Kuan** [1] , **Szu-Yung Wang** [2] **and Jiun-Hao Wang** [1,*]

1    Department of Bio-Industry Communication and Development, National Taiwan University,
     Taipei 10617, Taiwan; mikayla571@gmail.com
2    Department of International Business, National Taiwan University, Taipei 10617, Taiwan;
     d05724006@ntu.edu.tw
*    Correspondence: wangjh@ntu.edu.tw

**Abstract:** Agricultural product certifications have proliferated due to the growing concerns in many countries over food safety and environmental sustainability. Encouraging farmers to self-organize was regarded as a useful tool to enhance the adoption of agricultural product certifications in Taiwan. However, previous studies solely focused on the association between membership in a production organization and single food certifications. Moreover, little is known of how different types of organizational participation could improve the adoption of agricultural product certifications. This study seeks to fill this knowledge gap by employing multinomial logistic regression model (MNL) to investigate factors affecting farmers' decisions to participate in agricultural product certification. Special attention is paid to the role of different types of organizational participation in farmers' choices for agricultural product certification. The study used a nationally representative sample of core farmers in Taiwan, and its results revealed evident differences in farmers' organization types. For example, the farm operators who participate in agricultural cooperatives (co-ops) tend to use organic labels. In contrast, farm operators who participate in agricultural production and marketing groups (APMGs) tend to adopt the Traceable Agricultural Products (TAP) label. Moreover, age, education level, farming experience, farm labor, farm type, agricultural facilities, and regional location have a significant effect on farmers' choices for participating in agricultural product certification across different models. The findings suggest that policymakers should consider these differences in the organizational operation of APMGs and co-ops and provide customized measures by promoting different types of agricultural product certifications.

**Keywords:** farmers' organizations; agricultural product certifications; agricultural cooperatives; agricultural production and marketing group; Good Agricultural Practices (GAP); Traceable Agriculture Products (TAP)

## 1. Introduction

Agricultural product certification schemes for farmers have gained considerable attention since the early 1990s [1]. Additionally, there is an increasing concern worldwide about food safety, environmental sustainability, and farmers' rights (e.g., labor conditions, gender equality, and producer welfare) in the agricultural sector [2,3]. Thus, many voluntary certification standards involving farmers have been introduced as a key approach to solving these issues [4]. A series of agricultural product certification projects have been implemented in Taiwan since 1994, including Good Agricultural Practices (GAP) in 1994, Traceable Agriculture Products (TAP) in 2007, and organic agricultural products in 2009 [5–7]. Unfortunately, the adoption rate of farmers participating in the certification labeling system is still not as encouraging as expected [6,8]. Understanding the factors that

are correlated with farmers' decisions to participate in agricultural certification projects is crucial to effectively promoting these projects [3].

There has been a wealth of research on agricultural product certifications and the factors associated with their adoption [8–13]. The literature documents broad categories of factors influencing farmers' adoption decisions, including socio-demographic characteristics, farm size, farming experience, farm income, labor conditions, farm types, and farm location. However, the influence of these factors on farmers' adoption decisions varied across countries. For example, in some European countries and the US, a farmer's gender, education level, farming experience, family size, farm income, and organizational membership were important determinants regarding the adoption of organic systems [14–16]. Conversely, a study by Singh and Maharjan [17] in Nepal reveals that gender has no role in the adoption decisions of smallholders [13]. Likewise, farm size and farming experience positively influence decision making in certification adoption in Turkey [18], while those factors negatively impact organic certification in the US, Thailand, and West African contexts [15,19,20]. Understanding the region-specific determinants regarding the adoption of agricultural product certifications is crucial for the successful diffusion of sustainable agricultural practices in different countries. Against the above backdrop, this study attempts to assess the determinants of adopting three agricultural product certifications, including GAP, TAP, and organic agricultural products, in Taiwan.

According to the perspective of the Overseas Cooperative Development Council [21] in 2007, cooperative management was regarded as a useful means for organizing smallholders to overcome economic and market constraints by enhancing their collective bargaining power. This perspective implied that farmers' organizations increase the feasibility of agricultural product certification for small-scale farmers by developing economies of scale [22–24]. In this context, strengthening the farmer's organizational capacity is regarded as an approach to promoting agricultural product certification. After Taiwan joined the World Trade Organization in 2002, the agricultural sector was compelled to improve food safety and quality to enhance its competitiveness in the global agricultural market. Such organizational participation is especially important for encouraging farmers to adopt agricultural product certifications in Taiwan, where most farmers are small-scale holders with low production yields, aging labor, and weak market competition [25,26]. Although the organizational approach has attracted attention among agricultural economists, little is known about how the approach affects agricultural product certification choices. As is evident, there are limited empirical studies on such issues in Taiwan, which have focused solely on the association between specific organization membership and single food certifications [3,8,25].

Previous studies indicated that farmers' organizations have the assistance available to facilitate the adoption of agricultural product certifications in many countries [9,23,25]. For instance, Monteiro and Caswell [27] found that the adoption of traceability certifications among farmers was affected by their membership in particular producer organizations. Wollni and Andersson [28] observe that farmers with organizational membership are more likely to adopt organic certifications due to the organizations providing access to related information and assistance for the adoption decision. Snider et al. [29] found that farmers' organizations encourage the adoption of voluntary certifications through training farm management practices. However, the association between membership in agricultural groups and certification adoption is inconclusive [30]. Experts, such as Ssebunya et al. [31], highlight that with or without certification, long-standing group membership has positive income effects. This result implies that participating in organizations will not necessarily increase the possibility of farmers adopting labels. Ruben and Fort [32] also suggest that dissatisfaction with organizational service provision will reduce farmers' willingness to obtain agricultural certification. In addition, the existing literature presents substantial supporting evidence from membership of farmers' organizations, which includes training, information acquisition, contact with extension agents, access to resources and markets, technological support, motivation, and interaction with other members [33–35]. This

argument means that farmers' participation in these organizations may be influenced by multifaceted interests and will enhance incentives for group certification. Therefore, more research is required to explore the role of participation in farmers' organizations in certification adoption.

The previously mentioned literature tends to treat all farmers' organizations without distinction, ignoring their diverse roles and functions. However, the diversity of farmers' organizations might imply different organizational capacities, interests, and responses to agricultural certification [36]. For example, assets and financial capital affect an organization's ability to provide benefits, influencing its members' perceptions of certification [34,37,38]. Furthermore, Latynskiy and Berger [23] note that the adoption of group certification depends on the size of the farmers' organization. In this regard, for smaller and less efficient organizations, group certification has a lower income effect, and it becomes less profitable. In general, a solid organizational infrastructure and management capacity will likely pursue the successful implementation of group certification. Therefore, this study considers individual participation in different farmers' organizations and explores how organizational participation determines farmers' certification choices [23,39].

This paper aims to investigate the determinants of farmers' agricultural certification decisions. To this end, special attention was paid to the relationship between the different types of farmers' organizational participation and farmers' certification decisions. The following research problems regarding farmers' adoption decisions were addressed. First, what factors are associated with agricultural certification adoption behaviors? Second, to what extent do socio-demographic characteristics and farming factors affect the adoption of these certifications? Finally, how do two types of organizational participation influence the decision making of agricultural certification adoption? In contrast to existing studies on this topic, this study is unique in several ways. First, unlike earlier studies that relied on the collection of data from limited sample sizes or restricted areas [11,40], this study utilizes a nationwide representative survey of farm households in Taiwan, which allows for a larger-scale evaluation of the impacts of agricultural cooperatives. Second, this study distinguishes the effects of different organizational categories on agricultural product certification decisions. Due to inherent organizational features, different types of farmers' organizations cannot be analyzed or treated as homogeneous entities. Third, most previous studies have only considered the certification decision as a binary choice between "yes or no" [4,8,41]. This study goes one step further by defining the adoption decision of agricultural certification as multiple choices: no certification, GAP label, TAP label, and organic label. This study includes farmers participating in various farmers' organizations and different types of certifications, which allows for broader conclusions to be drawn.

The remainder of this paper is organized as follows. First, a brief introduction is provided on the background of the agricultural product certification system and farmers' organizations in Taiwan, followed by an explanation of the data. An empirical model is then presented, and the results are discussed. Finally, the paper concludes with a summary and discussion of policy implications.

## 2. Overview of the Agricultural Product Certification System in Taiwan

Along with the liberalization of world trade in the early 2000s, food safety has become one of the major concerns in the agricultural sector. The Council of Agriculture (COA) integrated a series of agricultural labeling systems to comprehensively promote safe agricultural policies in Taiwan. The development of agricultural certification systems is associated with meeting the needs of consumers that differ in scope and history. As exhibited in Table 1, the GAP label, launched in 1994, was the first certification label, followed by the TAP label in 2007 [5,6]. Organic certification became integrated into the agricultural labeling system in 2009.

The GAP certification aims to implement safe pesticide use methods, record pesticides employed, and enable consumers to recognize safe products. The application and review process for GAP certification is not applicable for individual farmers but only for agricul-

tural production and marketing groups (APMGs) [5]. The GAP certification only requires APMGs to send samples for pesticide residue inspection rather than on-site sampling inspection. In 2015, 2127 APMGs had passed the GAP inspection. Furthermore, the total area of the certified GAP accounted for 25,761 hectares of cultivated land [42]. Therefore, GAP was the first certification label but is less stringent than other labeling systems.

**Table 1.** Comparison of different agricultural product certifications in Taiwan.

| Categories | GAP Certification | TAP Certification | Organic Certification |
|---|---|---|---|
| Label |  |  |  |
| Year began | 1994 | 2007 | 2009 |
| Third-party certification | No | Yes | Yes |
| Certification fee | Free | Low | High |
| Legal basis | None | TGAP regulation | Organic Agriculture Promotion Act |
| QR code | By producer group | By production batch | None |
| Applicants | Group | Group or individual | Group or individual |
| Penalties for violations | No | Yes | Yes |
| Regulation features | Restrictions on agrochemical use, only safe application allowed | From production to sale, all stages must be recorded and traceable; allows safe application of agrochemicals | Prohibited use of pesticides, GMOs, and other agrochemicals; eco-friendly farming practices required |
| Price premium vs. non-certified use | Relatively high | Higher | Highest |

In addition, the TAP is regarded as the cornerstone for building the formal agricultural certification system per the "Agricultural Production and Certification Act". Compared with conventional agricultural practices, TAP-certified products with health labels and higher prices achieved explosive growth in farming area and production volume in recent years. The TAP certification focuses on safe, sustainable, and open information traceable products. The TAP certification system requires farmers to record a profile for each product, including details of inputs, farming work forms, production, processing, packaging, and transportation to sales [8]. The TAP certification is applicable for individual farmers, cooperatives, or other producer groups. The TAP system is promoted through implementing a TAP information platform and formulating operating standards, such as Taiwan's Good Agricultural Practices (TGAP). In 2015, there were 1570 certificated applicants. In the same year, 11,209 hectares of farmland had been certified, and an average of 5.27 million labels were used per month, which grew by 47% compared with the same quarter the year before [7]. Compared with the other two certification systems, the inspection of organic products was the most rigorous and environmentally friendly certification. Organic agricultural products are designed to reject the use of chemical fertilizers and pesticides. According to the record, 2598 farmers and 6490 hectares of farmland received organic certification in 2015 [42].

In addition to quality assurance and food safety guarantee, the GAP, TAP, and organic certifications also contribute to certifying the product origins, generating price premiums, and offering better marketing opportunities compared with conventional products [8,43]. The advantages of agricultural certifications can be summarized as follows: qualification based on nationally recognized standards, government subsidies, price premiums, and meeting the marketing requirements for superior market access. Conversely, lower produc-

tivity and yields, higher input costs, and expensive inspection fees have hindered farmers' willingness to adopt the certification labels.

### 3. The Background of Farmers' Organizations in Taiwan

The farmers' associations (FAs), agricultural cooperatives (co-ops), and agricultural production and marketing groups (APMGs) are three traditional farmers' organizations with a legal basis in Taiwan [44]. APMGs, which are sub-organizations under the FAs, are the most basic organizations for farmers. In 1952, several FAs established different grassroots groups for autonomous learning and cooperative activities to strengthen agricultural extension services. Considering that APMGs play an important role in expanding the scale of agricultural operations and enhancing social participation, the government integrated existing organizations collectively, redefined their organizational functions, and referred to agricultural production and marketing groups by the Agricultural Development Act in 2004. As a result, there were 6518 APMGs in 2014, and the share of crops groups accounted for 87.8% (i.e., 5725 APMGs) of the total figure [45]. Thus, agricultural authorities and extension agents could provide group counseling and assistance to APMGs in terms of operations management, production technologies, and marketing capabilities instead of individual advisory services.

An agricultural cooperative is defined as a cooperative established by natural persons engaged in agriculture, i.e., crop, livestock, fish farming, forestry, or any agribusiness-related service, based on their willingness and common need to cooperate. The agricultural cooperatives are regarded as a useful means for policy implementation toward small-scale agricultural development. The establishment of agricultural co-ops must align with Taiwan's Cooperative Act, which could be divided into different categories, such as agricultural production, transportation, marketing, co-ops, or both. In 2015, there were 1106 co-ops and 150,244 members in Taiwan [46].

The main differences between the two forms of farmers' organizations are that the APMGs are sub-organizations under local FAs, without legal person status. Besides, profits and losses of APMGs are borne by individual members, and the organization's convention is customized and quite flexible. As for agricultural cooperatives, in addition to the legal person status, farmers can freely apply for establishment in a series and share the profits and losses of cooperatives. A more detailed comparison of APMGs and co-ops is shown in Table 2.

**Table 2.** Organizational comparison of co-ops and APMGs in Taiwan.

| Category/Group | Agricultural Cooperatives (Co-Ops) | Agricultural Production and Marketing Groups (APMGs) |
| --- | --- | --- |
| Legal basis | Cooperatives Act, a national law | Regulations for Establishment and Guidance of Agricultural Production and Marketing Groups, an interior regulation of the COA |
| Competent authority | Ministry of the Interior | Council of Agriculture |
| Legal person | Yes | No |
| Organizational function | Single specific functional organization; based on the shared economic needs of members, such as production, transportation and marketing, supply, utilization, and labor | Responsible for handling joint agricultural procurement and implementing joint agricultural production and marketing plans |
| Administrative level | Two-level system: the national level and the county (city) level | Only at the township level can a subordinate organization belongs to the local farmers' association |
| Application requirements | Free to apply for establishment by farmers with common needs | 1. Farmers aged over 18 who have connected land or a common agricultural product; 2. Farmers can only participate in one group for each agricultural product |
| Profit and loss responsibility | Profits and losses are shared by all members | Profits and losses are borne by the individual |

## 4. Materials and Methods

### 4.1. Data

The data used in this study was drawn from the 2013 Core Farm Households Survey (CFHS), conducted by the Directorate-General of Budget, Accounting and Statistics (DG-BAS) in Taiwan. Core farm households are defined as farm households with an annual farm income of more than TWD 200,000, comprising at least one household member aged below 65, and engaged in full-time farming activities [47]. Therefore, the CFHS dataset consists of a nationally representative sample of full-time farmers. Data were collected from selected farm operators using standardized face-to-face interviews by local agricultural officers, and a multistage stratified sampling scheme was used to select a probability sample. In total, 9951 core farm households were interviewed. The survey's primary focus was to understand farm production characteristics and the socio-economic information of core farm households. Farm labor and agricultural product sales data were also documented. Given that this study's primary focus was to understand the association between organizational participation and agricultural product certification, respondents who participated in farmers' organizations, e.g., co-ops and APMGs, were selected for further analysis. After excluding observations with missing values, 3853 core farm households are included in the empirical analysis. The proportion of farmers' self-organized groups is about 38.7%, reflecting the significance of farmers' organizations in Taiwan [25].

### 4.2. Measurements

The information contained in the CFHS dataset includes farmers' characteristics, farm features, labor conditions, the status of organization participation, and the adoption of agricultural product certification. All nominal variables were treated as dummy variables in the analytical models, while other numerical indicators were treated as continuous measures. Firstly, this study uses different types of agricultural product certifications to measure the sustainability of farm production. It defines several dummy variables in relation to agricultural certifications if the agricultural product has been certified in compliance with the Agricultural Production and Certification Act [48], including the GAP label (=1), TAP label (=1), and organic label (=1). Regarding farmers' cooperative organizations, the study creates binary variables to define organizational participation in co-ops (=1) or APMGs (=1).

The explanatory variables included in this study's analysis are built on the empirical specifications from previous studies. Consistent with the specification used in studies by Jiang and Yir-Hueih [3], Liao, Chang, and Chang [8], Pradhan, Tripura, Mondal, Darnnel, and Murasing [10], Azam and Banumathi [16], Aidoo and Fromm [19], and Monteiro and Caswell [27], the number of selected variables was specified. The selected variables that may be associated with farmers' choices of agricultural product certifications in this analysis were classified according to three dimensions: farmer characteristics (i.e., gender, age, and level of education), farm and farming factors (i.e., farm size, farming experience, farm income, labor conditions, farm types, and others), and regional location. In addition, several variables reflecting organizational participation, socio-demographic characteristics, farm features, labor conditions, and regional heterogeneity are hypothesized as being associated with agricultural product certification.

These variables include age (in years), gender (male = 1), and educational attainment of the farm operator (e.g., primary or below, junior high, senior high, and college or above). Another continuous variable accounting for farming experience is also specified. Regarding farm laborers' features, this study defined several continuous variables to measure the number and structure of laborers working on the farm, including household members in addition to regular and temporary workers. In addition, the size of farmland is also captured using another continuous variable. A continuous variable representing the total farm revenue of agricultural products for sale was also defined (in TWD 1 million). Several dummy variables for different farm types were also included: food crop farms (=1), high-value crop farms (=1), and livestock farms (=1). Moreover, a dummy variable was

specified to indicate whether a farm operator used agricultural facilities on their farm (e.g., greenhouses, net rooms, animal sheds, or other agricultural facilities) (if yes, = 1). Finally, the study controlled for regional heterogeneity by using four geographic dummy variables: northern, central, southern, and eastern regions (=1).

In considering the empirical model's potential endogeneity, the agricultural certification decision may be correlated with unobserved factors that also influence the organizational participation of farmers, in which case the estimated effect would suffer from selection bias. Therefore, the study uses an instrumental variable (IV) approach to deal with the potential endogeneity problem. One shortcoming of the IV approach is that the analysis builds on the existing dataset (i.e., the CFHS), so not all possible unobserved biases may have been eliminated. According to the CFHS questionnaire, the transfer payments include agricultural subsidies, old-age farmers' welfare allowances, agricultural disaster relief payments, or scholarships for farmers' children from the agricultural authority. The eligibility of most transfer payments requires membership in the farmers' organization rather than the co-ops membership. Therefore, this study used the government's transfer payments as a valid instrument for farmers' certification decisions to reduce potential endogeneity. Detailed operational definitions and measurements for the selected variables are defined in Table 3.

*4.3. Hypothesis*

Based on the literature referred to above [9,14], several factors are expected to influence the implementation of farmers' certification decisions. Specifically, the role of farmers' organizational participation in influencing farmers' choices regarding certifications has been less documented in Taiwan. Thus, the following research hypotheses will be addressed:

**Hypothesis 1.** *The farmer's decision to participate in agricultural certification is affected by their socio-demographic characteristics, farm features, and regional location.*

**Hypothesis 2.** *The effects of the previously mentioned factors on the farmer's decision to participate in different agricultural product certifications are varied.*

**Hypothesis 3.** *The different types of farmers' organizational participation engender varying choices on the adoption of agricultural product certification.*

*4.4. Statistical Analysis*

The study's statistical analysis involved several steps. The first part presents the personal characteristics, starting with the descriptive statistics of the full sample. Next, the study examined the organizational participation of farm operators using the chi-square test and *t*-test to compare the distribution of socio-demographic characteristics, farm features, farm labor, regional location, and agricultural product certification decisions between those who participated in co-ops and APMGs. In the final stage, the study applied a multinomial logistic regression model (MNL) to estimate the relationship between organizational participation factors and the likelihood of having different types of agricultural product certifications.

The MNL model is the most widely used model due to its easy estimation and stability. However, it imposes the restrictive assumption that choices are independent across alternatives. Since another multinomial probit model (MNP) does not impose the independence assumption, one might reasonably argue that the MNP is a better model. However, these concerns are exaggerated, and, under most circumstances, MNL estimation performs as well or better than the MNP [49]. Following the literature on discrete choice models, the multiple choices for agricultural product certifications are specified in the MNL model. The study defines a discrete choice variable ($j$) whose value is "0" for farms with no certification, "1" for farms that adopt the GAP label, "2" for farms that adopt the TAP label,

and "3" for farms that adopt the organic label. The probability of each farmer's choice for certification can be specified as follows:

$$P_j = \frac{\exp(X'\beta_j)}{\sum_{j=0}^{3} \exp(X'\beta_j)} \ , \ j = 0, \ 1, \ 2, \ 3 \tag{1}$$

**Table 3.** Selected variables' definitions and sample statistics (N = 3853).

| Selected Variables | Definition and Measurement | Mean (%) | SD (Freq.) |
|---|---|---|---|
| *Agricultural product certification types* | | | |
| No certification | If the farm hasn't achieved any certification (=1). | 0.68 | 2614 |
| GAP label | If the farm has achieved GAP certification (=1). | 0.18 | 698 |
| TAP label | If the farm has achieved TAP certification (=1). | 0.10 | 387 |
| Organic label | If the farm has achieved organic certification (=1). | 0.04 | 154 |
| *Organizational participation* | | | |
| Co-ops | If the farm operator participated in agricultural cooperatives (=1). | 0.14 | 545 |
| APMGs | If the farm operator participated in agricultural production and marketing groups (=1). | 0.86 | 3308 |
| *Socio-demographic characteristics* | | | |
| Male | If the farm operator is male (=1). | 0.92 | 3545 |
| Age | Age of the farm operator (in years). | 56.66 | 10.52 |
| Educational level | | | |
| Primary or below | If the farm operator has completed primary education or below (=1). | 0.35 | 1349 |
| Junior high | If the farm operator has completed junior high school (=1). | 0.28 | 1079 |
| Senior high | If farm operator has completed senior high school (=1). | 0.32 | 1233 |
| College or above | If farm operator has completed college or higher education (=1). | 0.05 | 193 |
| *Farm features and labor conditions* | | | |
| Farming experience | Years the farm operator worked on the farm (in years). | 29.63 | 14.28 |
| Farm size | Operated farmland size (in hectares). | 1.58 | 1.88 |
| Farm labor (persons) | | | |
| Household members | Number of household members that worked on the farm. | 2.65 | 1.12 |
| Regular workers | Number of regularly hired workers that worked on the farm. | 0.23 | 1.67 |
| Temporary workers | Number of temporary workers that worked on the farm. | 2.98 | 5.80 |
| Farm type | | | |
| Food crops farm | If main product of farm is rice or grains (=1). | 0.32 | 1233 |
| High-value crops | If main product of farm is fruits, vegetable, flowers, mushrooms, or special crops (=1). | 0.62 | 2389 |
| Livestock farm | If main product of farm is animal products (=1). | 0.06 | 231 |
| Farm revenue | Annual revenue of farm production (in TWD 1 million). | 1.97 | 5.60 |
| Agricultural facilities | If the farm uses greenhouse, net room, animal shed or other agricultural facilities (=1). | 0.35 | 1349 |
| *Regional location* | | | |
| North | If farm located in the northern region (=1). | 0.14 | 539 |
| West | If farm located in western area (=1). | 0.50 | 1927 |
| South | If farm located in southern region (=1). | 0.29 | 1117 |
| East | If farm located in the eastern region (=1). | 0.07 | 270 |
| *Instrumental variable* | | | |
| Transfer payment | If the farm household has received agricultural subsidies, allowances, awards, or scholarships from the government (=1). | 0.83 | 3198 |

Note: 3853 farm operators are included. Abbreviation: SD, standard deviation.

In Equation (1), where $P_j$ is the probability when the farm operator chooses the $j$th type of agricultural certification, $X'$ composes a vector of explanatory variables, and the parameters to be estimated are $\beta_j$. This model is estimated by the maximum likelihood estimation procedure. This study is particularly interested in the coefficient $\alpha_1$ to evaluate whether organizational participation impacts agricultural product certification. However, one problem in estimating Equation (2) is that $APMGs_i$ is likely endogenous, that is, certification decisions might influence participation in co-ops or APMGs. The study addresses the endogeneity issue by selecting an instrumental variable to estimate the relationship between the $APMGs_i$ membership and explanatory variables. The linear equation system is specified as:

$$
\begin{aligned}
APMGs &= \beta_1 z_i + \beta_2 X_i + \varepsilon_{1i} \\
certification_i &= \alpha_0 + \alpha_1 APMGs_i + \alpha_2 X_i + \varepsilon_{2i}
\end{aligned}
\tag{2}
$$

where $certificate_i$ is the $i$th farm operators' choice of agricultural certification ($certification = 0$ if an operator participates in no certification, $certification = 1$ if GAP label, $certification = 2$ if TAP label, and $certification = 3$ if organic label), $X_i$ is a vector of socio-demographic characteristics, farm features, farm labor, and regional location variables listed in Table 3, $z_i$ is the transfer payment used as an instrumental variable for participating in the $APMGs$, and $\varepsilon_{1i}$ is a normally distributed error term. The consistent estimates $\alpha_0$, $\alpha_1$, $\alpha_2$, $\beta_1$, and $\beta_2$ can be obtained by using the two-stage least squares regression method (2SLS), and the validation of the selected instruments are reported in Appendix A. In addition, the multicollinearity was also assessed using the variance inflation factor (VIF) and no problem was detected (all VIF values were below 10).

## 5. Results

### 5.1. Descriptive Statistics of the Sample Characteristics

Table 3 presents the sample statistics of the selected variables. Among the 3853 farm operators, 68% did not adopt any agricultural product certification. The socio-demographic characteristics of the sample show that 92% of respondents were male. In addition, the features of farm operators showed that the average years of farming experience was 29.6 years, the average farm size was 1.58 hectares, the average age was 56.7 years, and most of them had graduated from senior high school or lower, whereas only 5% of them had completed college-level education or higher. The average number of household members working on the farm was 2.65. The average farm income was TWD 1.97 million per year, 62% tend to produce high-value crops (including fruits, flowers, mushrooms, or special crops) and 35% have used agricultural facilities. For the regional characteristics, about 58% of the sample reported living in the western area.

### 5.2. Association between Farmers' Association Participation and Selected Variables

Table 4 presents the distribution of the socio-demographic characteristics for farm operators who participated in co-ops and APMGs. Among the 3853 farm operators, 545 (14%) are members of co-ops and 3308 (86%) participate in APMGs. In general, compared with APMG members, farmers who engage with co-ops are more likely to be male, younger, possess higher education, or be those living in western areas. In addition, farmers who are members of co-ops are more profitable and have larger farms. On average, farmers who engage with co-ops observe farm profits of TWD 3.79 million, and the average farm size of participants is 2.09 hectares. Participants in co-ops also have a higher number of laborers (including household members, regular workers, and temporary workers) working on the farm, tend to produce livestock farms, and use agricultural facilities. The results showed that the differences in the most selected socio-demographic characteristics and organizational participation were statistically significant, except for educational level, farming experience, and farm size.

**Table 4.** Group comparison between co-op and APMG participation.

| Selected Variable | Co-Ops (N = 545) | | APMGs (N = 3308) | | X²/t-Value |
|---|---|---|---|---|---|
| | Mean (%) | SD (Freq.) | Mean (%) | SD (Freq.) | |
| Male (=1) | 0.94 | 512 | 0.91 | 3010 | 7.27 *** |
| Age (years) | 55.93 | 9.90 | 56.78 | 10.62 | 1.75 * |
| Educational level | | | | | |
| Primary or below | 0.32 | 174 | 0.35 | 1158 | |
| Junior high | 0.30 | 164 | 0.28 | 926 | 2.90 |
| Senior high | 0.33 | 180 | 0.32 | 1059 | |
| College or above | 0.06 | 33 | 0.05 | 165 | |
| Farming experience (years) | 29.28 | 13.22 | 29.69 | 14.45 | 0.62 |
| Farm size (hectare) | 2.09 | 3.01 | 1.93 | 2.78 | 1.23 |
| Farm labor (persons) | | | | | |
| Household members | 2.73 | 1.25 | 2.64 | 1.10 | 1.73 * |
| Regular workers | 0.35 | 3.16 | 0.21 | 1.27 | 1.81 * |
| Temporary workers | 5.12 | 9.35 | 2.62 | 4.89 | 9.43 *** |
| Farm type | | | | | |
| Food crops farm | 0.27 | 147 | 0.32 | 1059 | |
| High-value crops farm | 0.56 | 305 | 0.64 | 2117 | 149.71 *** |
| Livestock farm | 0.18 | 98 | 0.04 | 132 | |
| Farm revenue (TWD million) | 3.79 | 8.44 | 1.67 | 4.92 | 8.26 *** |
| Agricultural facilities | 0.40 | 218 | 0.34 | 1125 | 7.03 *** |
| Regional location | | | | | |
| North | 0.10 | 55 | 0.15 | 496 | |
| West | 0.58 | 316 | 0.48 | 1588 | 26.88 *** |
| South | 0.28 | 153 | 0.29 | 959 | |
| East | 0.04 | 22 | 0.08 | 265 | |

Note: 3853 farm operators are included. *** and * indicate the significance at the 1%, and 10% level, respectively. Abbreviation: SD, standard deviation.

### 5.3. Association between Organizational Participation and Agricultural Product Certification

As seen in Table 5, the farmers participating in co-ops are less likely to adopt agricultural product certifications than their APMGs counterparts. On average, up to 75.96% of the farmers engaging with co-ops have not obtained agricultural product certification—more than those in APMGs (66.51%). Similarly, a lower ratio of co-op participants has also adopted the GAP and TAP labels: 9.17% and 8.99%, respectively. In contrast, farms participating in co-ops are more likely (5.87%) to use the organic label than their counterparts (3.69%). In general, these certifications have not proven as successful as expected since a lower participation rate has been recorded.

### 5.4. The Determinants Associated with the Choice of Agricultural Product Certification

The factors influencing the adoption of agricultural product certifications were examined using the MNL model. Among the different types of agricultural product certifications, the study chose "no certification" as the reference group since it represents the highest proportion (68% of all certification choices). Therefore, all the estimated coefficients for the selected variables represent the effect of the variables on the specific certification compared with the "no certification" choice.

**Table 5.** Association between organizational participation and agricultural product certifications.

| Types | Co-Ops (N = 545) | | APMGs (N = 3308) | | $X^2$ |
| --- | --- | --- | --- | --- | --- |
| | Freq. | % | Freq. | % | |
| No certification | 414 | 75.96 | 2200 | 66.51 | |
| GAP label | 50 | 9.17 | 648 | 19.59 | |
| TAP label | 49 | 8.99 | 338 | 10.22 | 40.47 *** |
| Organic label | 32 | 5.87 | 122 | 3.69 | |

Note: 3853 farm operators are included. *** indicate the significance at the 1% level.

Table 6 displays the estimations of the several logistic regression analyses, which include coefficients, standard errors, odds ratios (i.e., Exp($\beta$)), and significance levels. The study began by looking at the findings of the statistical tests (bottom of Table 6). For the likelihood ratio test, the log pseudo-likelihood value is $-3322.78$, which is higher than the critical value at the 1% level ($p < 0.001$). Therefore, the null hypothesis that all slope coefficients are zero was rejected. In general, the farmers' decisions on agricultural certification were significantly influenced by organizational participation, socio-economic characteristics, farm production features, and regional locations. However, the effects and statistical significance of each explanation vary inconsistently across different agricultural certification models.

In the GAP model, the respondents who were younger, had more farming experience, engaged in food crop farming activities using agricultural facilities, and were located in the southern and eastern regions were more likely to participate in the GAP certification than their counterparts. For example, the odds ratio of increasing one year of farming experience is more likely to adopt the GAP certification than "no certification" by a factor of 1.01, given that all other variables remain constant. In addition, compared with their counterparts, the odds ratios of those farms located in southern and eastern regions for adopting the GAP certification are 1.53 and 1.75, respectively (compared with "no certification"). However, there is no significant association found between organizational membership and GAP certification. A more detailed discussion will be provided in the follow-up section.

Although similar patterns were found across the other two logistic models' estimations, significant differences existed. In the TAP model, organizational participation, age, farming experience, the number of temporary workers, high-value crop and livestock farms, agricultural facilities, and regional location were significantly associated with an increase in the proportional odds of adopting the TAP certification. The respondents who participated as members in co-ops were less likely to adopt the TAP certification, compared with those that were members of APMGs, by 64%. In addition, a higher likelihood of adopting the TAP certification, compared with "no certification", is evident for longer farming experience, more temporary workers, high-value crop farms (vs. food crop farms), and agricultural facilities used. In this regard, the odds ratios are 1.01, 1.02, 1.30, and 1.43, respectively.

The certification model of organic products indicated that co-op members are more likely to adopt the organic certification than APMG members. The odds ratio accounts for 1.14. Moreover, higher educational level, more farming experience, farm revenue, and used agricultural facilities are positively associated with adopting the organic certification. Therefore, these respondents have a higher probability of participating in the organic certification program. In summary, although the effects of explanatory variables on different agricultural certifications vary across different models, the direction and significance of relevant determinants remained largely unchanged.

**Table 6.** Estimation results of multinomial logistic model for GAP, TAP, and organic labels (N = 3853).

| Selected Variables | GAP Label | | | | TAP Label | | | | Organic Label | | | |
|---|---|---|---|---|---|---|---|---|---|---|---|---|
| | β | | Exp(β) | SE | β | | Exp(β) | SE | β | | Exp(β) | SE |
| Co-ops (Ref.= APMGs) | 0.94 | | 2.57 | 0.41 | −0.45 | *** | 0.64 | 0.08 | 0.24 | ** | 1.14 | 0.06 |
| Male (=1) | −0.06 | | 0.94 | 0.11 | −0.01 | | 0.99 | 0.12 | −0.29 | | 0.54 | 0.22 |
| Age (years) | −0.01 | ** | 0.99 | 0.00 | −0.01 | ** | 0.99 | 0.01 | −0.01 | | 0.97 | 0.01 |
| Educational level (Ref.= primary or below) | | | | | | | | | | | | |
| Junior high | −0.13 | | 0.88 | 0.09 | −0.10 | | 0.91 | 0.09 | 0.19 | | 1.42 | 0.12 |
| Senior high | 0.00 | | 1.00 | 0.10 | 0.04 | | 1.04 | 0.11 | 0.29 | ** | 1.75 | 0.15 |
| College or above | 0.03 | | 1.03 | 0.16 | 0.08 | | 1.09 | 0.18 | 0.45 | * | 2.64 | 0.25 |
| Farming experience (years) | 0.01 | *** | 1.01 | 0.00 | 0.01 | *** | 1.01 | 0.00 | 0.02 | * | 1.05 | 0.01 |
| Farm size (hectare) | 0.01 | | 1.01 | 0.01 | 0.01 | | 1.01 | 0.01 | 0.01 | | 1.03 | 0.01 |
| Farm labor (persons) | | | | | | | | | | | | |
| Household members | −0.02 | | 0.98 | 0.03 | −0.01 | | 0.99 | 0.03 | 0.06 | | 1.15 | 0.04 |
| Regular workers | 0.01 | | 1.01 | 0.02 | −0.01 | | 0.99 | 0.03 | 0.00 | | 1.02 | 0.02 |
| Temporary workers | 0.01 | | 1.01 | 0.01 | 0.02 | ** | 1.02 | 0.01 | 0.00 | | 1.01 | 0.01 |
| Farm type (Ref.= food crops farm) | | | | | | | | | | | | |
| High-value crops farm | 0.12 | | 1.13 | 0.08 | 0.26 | * | 1.30 | 0.14 | −0.21 | | 0.68 | 0.14 |
| Livestock farm | −1.43 | *** | 0.24 | 0.25 | −1.13 | *** | 0.32 | 0.28 | 0.55 | * | 2.29 | 0.29 |
| Farm revenue (TWD million) | 0.00 | | 1.00 | 0.00 | 0.00 | | 1.00 | 0.00 | 0.00 | * | 1.02 | 0.00 |
| Agricultural facilities (=1) | 0.40 | *** | 1.49 | 0.07 | 0.36 | *** | 1.43 | 0.09 | 0.08 | | 1.30 | 0.12 |
| Regional location (Ref.= north) | | | | | | | | | | | | |
| West | 0.14 | | 1.15 | 0.10 | 0.12 | | 1.13 | 0.11 | −0.91 | *** | 0.16 | 0.42 |
| South | 0.42 | *** | 1.53 | 0.10 | 0.51 | *** | 1.67 | 0.17 | −0.88 | *** | 0.18 | 0.41 |
| East | 0.56 | *** | 1.75 | 0.14 | 0.60 | *** | 1.83 | 0.21 | −0.40 | | 0.53 | 0.25 |
| Intercept | −0.93 | *** | 0.39 | 0.28 | −1.03 | *** | 0.36 | 0.59 | −1.31 | ** | 0.18 | 0.47 |
| Observation | | | | | | 3853 | | | | | | |
| Log pseudo-likelihood | | | | | | −3322.78 | | | | | | |
| Pseudo R² | | | | | | 0.07 | | | | | | |

Note: No certification as the reference category. The standard errors are cluster in cooperative. ***, **, and * indicate the significance at the 1%, 5%, and 10% level, respectively. Abbreviation: SE, standard error. Dummy of transfer payment is used as an instrument for co-ops.

## 6. Discussion

The results reveal a significant relationship between farmers' organizational participation and the choice of agricultural product certification, except for the GAP label. As hypothesized, organizational membership had a significant effect, implying that the farmers affiliated with an agricultural group or organization were more likely to adopt TAP and organic labels [28]. This result reflects that the activities in groups and knowledge shared by other farmers help the farmers to access information easily and solve problems regarding group certification and group marketing [28,31]. This study's finding is supported by previous studies documenting the significant and positive influence of group membership on farmers' adoption behaviors [27,29].

Regarding different effects of organizational participation, the results indicate that farm operators participating in APMGs are more likely to use the TAP labels than their counterparts in co-ops. In contrast, the farm operators who participated in cooperatives are more likely to use organic label certifications. This finding supports the hypothesis that different types of farmers' organizational participation influence varying choices of

agricultural certification labels. The result is plausible that cooperative membership is more likely to have more production advantages for adopting organic label certifications due to their capacity to create economies of scale [23,34]. In addition, co-ops operating in the form of enterprises can have large-scale operations, providing greater motivation and possibilities for capital investment to enhance product quality and food safety, and better market access (i.e., supermarket and wholesale chains) as compared with individual producers [3,50]. This finding means that farmers with cooperative membership have more resources and enjoy more advantages (e.g., lower production cost, efficient farm management, better access to markets and supplies, more policy-induced subsidies), making them dedicated to adopting organic farming practices [34,51]. Although cooperatives can get a great advantages in knowledge acquisition or government subsidies, the adoption of organic certification has still been slow in Taiwan. Not every cooperative member is comfortable transitioning because of the associated barriers. These barriers include strict inspection mechanisms, pesticide and agrochemical exclusions, high management costs, risks of above a two-year transition, amplified recordkeeping requirements, long-term field management and diverse crop rotations, land access securing, lack of markets to capture marketing premiums, and changes in the government's subsidy policy [52].

Most socio-demographic characteristics of farm operators, except gender, affect the selection of agricultural product certification. Farmers' ages influence their decisions to participate in agricultural product certification. In this regard, older farmers are less likely to choose the GAP and TAP label certifications compared with their younger counterparts. This finding may reflect the fact that older farmers are more risk-averse and less willing to utilize new farming practices (i.e., traceable techniques) than younger farmers [27]. This result is inconsistent with the smallholder cases in the Netherlands [16] and Nepal [12], which showed that older farmers had more opportunities for adoption than younger farmers. These outcomes may be due to a good relationship with an extension service and more experience in farming. The educational attainment of the farm operator is also a significant factor in the organic certification model. Results show that farm operators who have finished senior high school and above are more likely to have organic certification compared with their counterparts with primary education. This result aligns with the cases of small US farms [15] and smallholder farmers in Thailand [20], possibly reflecting that regardless of the regional difference, educated smallholder farmers are likely to be more attracted to the positive environmental externalities that organic farming practices generate.

Farming features also impact the decision toward farmers' agricultural product certifications. Farm operators who are more experienced in production tend to adopt the GAP and TAP labels as well as organic certification. This result may support the belief that farmers with more experience is usually older and less educated to shift to relatively new concepts of farming [20,27]. A reasonable explanation is that farmers with more farming experience may have a better understanding of farming practices and can evaluate the viability of participating in agricultural certification programs [10]. As expected, having more temporary workers working on the farm also increases the likelihood of obtaining TAP certification. This result aligns with the previous findings, which indicate that agricultural-certificated farming practices require more labor inputs than non-certificated ones [53]. However, no significant association is found between farm size, farm labor, and GAP as well as organic label certification.

Differences in participation across farm types are also evident. Compared to their counterparts, the high-value crop producers, producing crops such as vegetables, fruit, flowers, mushrooms, or special crops, are more likely to be involved in TAP labeling rather than GAP and organic labels. The result is plausible—fresh fruits and vegetables have become the main product promoted by the traceability system in many countries to reduce fresh food waste caused by perishability [54]. It may, therefore, be reflected that authorities targeted fruits and vegetables which are more easily traceable as priority projects during the initial stage of the TAP program in Taiwan [8,13,43]. The results also indicate that livestock farms are less likely to select the GAP and TAP labels compared with the high-value crop

producers. In contrast, livestock farm operators are more likely to adopt organic labels than food crop farmers. Therefore, it was logical to conclude that organic livestock products are gradually attracting consumers' attention and favor in Taiwan, and thus have a higher price premium than plant food products [55,56].

Furthermore, farm operators who use agricultural facilities are more likely to participate in the GAP and TAP certification systems than their counterparts. It is reasonable to expect that using agricultural facilities allows for better control of crop production, recording of environmental conditions, and meeting certification requirements [57]. The study's result, which was anticipated, suggests that farm operators who have more farm revenue are more likely to have organic certification. Consistent with the findings in Pradhan, Tripura, Mondal, Darnnel, and Murasing [10], this study discovered that higher incomes afforded farmers financial security, thereby supporting them in accepting the risks of investing in high-cost organic farming.

Finally, farm location has been shown to influence the decision to be certified. In this regard, those farmers living in the middle, south, and eastern regions of Taiwan have a greater likelihood of choosing the GAP and TAP labels. In contrast, farm operators located in the northern region are more likely to choose organic certification. One explanation for this is producers in the northern regions are responding to urban customers who may be more likely to demand organic products than consumers in other regions. As Hsu et al. [13] indicate, there is a higher concentration of organic food stores in the northern region, implying that there is strong demand for organic-certificated products in this region.

In all, farmers' decisions on agricultural certification were significantly influenced by socio-economic characteristics, farm production features, and regional locations. In this regard, the likelihood of adopting organic certification was observed among the well-educated (i.e., those with a senior high school education or above), those with a longer farming experience, farmers running livestock farms, those earning more farm revenue, and farms located in the northern region. Alternatively, farmers with more farming experience and temporary workers with high-value crop farms, those accessing agricultural facilities, or who lived in non-northern areas were more likely to choose TAP labels than non-certification. In addition, the target that is potentially more likely to adopt the GAP label is those with more farming experience, with food crop farms (compared to livestock farms), and those living in southern and eastern areas (compared to northern areas). As hypothesized, these results indicate that the factors affecting farmers' certification choices across farmers' organizations are different and should be considered in efforts to promote different agricultural certification practices in Taiwan.

## 7. Conclusions

This study used a national survey of core farm households to investigate the determinants of farm operators' participation decisions regarding agricultural certification programs. Special attention was accordingly paid to understanding the roles of the different types of farmers' organizational participation in certification decisions among the farmers. By estimating MNL model, several findings were revealed. First, the farmers' decisions for certification were significantly influenced by socio-economic characteristics, farm production features, and regional location [9,10,13]. Furthermore, it was evident that participation in farmers' organizations has significantly influenced decision making on the adoption of agricultural product certifications, except for the GAP label. The farm operators who participate in co-ops or APMGs make different choices on agricultural products certification labels. In this regard, the former is more likely to adopt organic labels while the latter tends to adopt TAP labels. This result implies that farmers' organizational participation increases the likelihood of adopting distinct agricultural certifications, providing evidence for the importance of access to information and assistance for the adoption decision [28,29].

This study goes beyond the existing literature on the topic in several ways [3,40]. First, its strength lies in its use of data from a large national survey examining the state of farmers'

organizational participation and the adoption rate of the agricultural product certification program in Taiwan. Second, as far as is evident, few studies have examined the association between farmers' organizational participation and agricultural product certification in Taiwan. These studies focused solely on the association between the whole membership of farmers' organizations and specific agricultural certification adoption [8,25]. Therefore, this study seeks to contribute to the existing literature on the topic in Taiwan; it further distinguished the effects of different farmers' organization categories on the different choices of agricultural certification, allowing for insight into the determinants of designated agricultural product certifications. Furthermore, examining the different effects of organizational characteristics allows for an understanding of how farmers' organizations can be more effective in improving the adoption of agricultural product certification [33,34]. Moreover, the study employed a 2SLS regression model to address the potential endogeneity of organizational participation in relation to their decision making on agricultural certification.

Several policy implications can be inferred from this study's findings. Given that the formation of farmers' organizations is necessary for sharing information and promoting agricultural product certification systems, more incentives must be established to drive farmers' participation. The incentive programs offered should be beyond economic incentives, such as subsidizing certification costs. Simplifying administrative procedures, providing sustainable agri-food training courses, or supporting ICT-based services also matter for facilitating agricultural certification. Furthermore, the results also show that up to 68% of the members of farmers' organizations have not adopted any agricultural certification, which means that the farmers' groups have limited support in the adoption of agricultural product certifications. Farmers who are already marginalized because of older age, poor education, limited financial capacity, land access, and lack of market accessibility may require additional support measures to improve their capacities, skills, and resources before they are able to benefit from membership in farmers' organizations.

The participation of different farmers' organizations could affect the adoption of different certifications for farmers. Policy or program prescriptions should consider these differences in the operation of APMGs and co-ops and provide targeted subsidies or tailor-made measures. The authorities must expand and coach the APMGs to facilitate their transformation into agricultural cooperatives to increase the adoption of organic certification. Such action can take the form of horizontally linking other groups to jointly form a cooperative. Compared with organic labeling, GAP and TAP certifications have lower thresholds and are easier to be accepted by farmers who implement conventional agriculture. Therefore, the government should encourage small-scale farmers to engage in production and marketing groups to increase the participation rate of the GAP and TAP programs. This study provides a preliminary understanding of the relationship between different types of farmers' organizations and agricultural certification adoption in a small-scale farming economic context. Even though officially certified food is likely to contribute to improving product competitiveness and farm income, it is acknowledged that smallholdings with limited available resources are challenged in qualifying under a complicated agricultural certification system. Thus, the findings of this study on Taiwan's agricultural product certifications could provide useful insights and serve as a reference to other smallholder countries [17,20].

Finally, some limitations of this study will be mentioned. In the first instance, the study's data were cross-sectional, preventing consideration of the dynamic aspects of farmers' participation in organizations and their agricultural product certification decisions. In addition, the data structure of the CFHS did not permit the research to further distinguish whether the farmer organization is a production, sales, purchase, supply, service, or other type of cooperative. Further detailed information (e.g., organization size, financial resources, the extent of farmer participation, or technical assistance capacity) related to farmers in organizational participation could be helpful in showing the robustness of these findings.

**Author Contributions:** Conceptualization, M.-Y.K. and J.-H.W.; methodology, data curation and statistical analysis, S.-Y.W.; writing, review and editing, M.-Y.K. and J.-H.W. The first two authors M.-Y.K. and S.-Y.W. contributed equally to this paper. All authors have read and agreed to the published version of the manuscript.

**Funding:** This research received no external funding.

**Institutional Review Board Statement:** Not applicable.

**Informed Consent Statement:** Not applicable.

**Data Availability Statement:** Core Farm Households Survey, 2013 (AA330001). Available from Survey Research Data Archive, Academia Sinica. doi:10.6141/TW-SRDA-AA330001-1.

**Conflicts of Interest:** The authors declare no conflict of interest.

## Appendix A. Validation of Selected Instrumental Variable

We report some statistical evidence to support our instrumental variable which is shown in the following table. Following Kleibergen and Paap [58] and Staiger and Stock [59], we have to examine whether the instrumental variable (i.e., transfer payment) passes the underidentification test and ensure the F statistic is greater than 10. The results indicate that the chi-squared value is 9.471 ($p$-value = 0.0021) and the F statistic is 10.447, which rejects the null hypothesis that the instrumental variable (transfer payment) is underidentified and jointly equal to zero.

| Variable | Coefficient | | SE |
|---|---|---|---|
| IV: transfer payment (=1) | 0.045 | ** | 0.009 |
| Male (=1) | 0.039 | ** | 0.011 |
| Age (years) | −0.001 | | 0.001 |
| Junior high | −0.002 | | 0.019 |
| Senior high | −0.020 | | 0.014 |
| College or above | −0.024 | | 0.022 |
| Farming experience (years) | 0.000 | | 0.001 |
| Farm size (hectare) | 0.000 | | 0.005 |
| Household members | 0.007 | *** | 0.001 |
| Regular workers | 0.004 | *** | 0.001 |
| Temporary workers | 0.010 | *** | 0.002 |
| High-value crops farm | −0.017 | | 0.010 |
| Livestock farm | 0.285 | ** | 0.076 |
| Farm revenue (TWD million) | 0.000 | | 0.000 |
| Agricultural facilities (=1) | −0.017 | | 0.026 |
| West | 0.058 | *** | 0.008 |
| South | 0.038 | * | 0.015 |
| East | −0.018 | | 0.020 |
| Intercept | 0.048 | | 0.022 |
| Statistical test | | | |
| Underidentification test | 9.471 | | |
| Weak IV test | 10.447 | | |

***, **, and * indicate the significance at the 1%, 5%, and 10% level, respectively. Abbreviation: SE, standard error.

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
