# Peer review of "Investigating the Association between Farmers’ Organizational Participation and Types of Agricultural Product Certifications: Empirical Evidence from a National Farm Households Survey in Taiwan"

_sustainability, doi:10.3390/su13169429_

Round 1

Reviewer 1 Report

Research problem should be more clearly defined.

The scientific problem and scientific novelty of the research should be emphasized more clearly in the abstract of the paper.

”Materials and methodes ” part should present the research questions, methods and hypothesis.

The results should be presented in the context of the existing findings.

Conclusions should more extensively juxtapose with literaturÄ™. 

Reviewer 2 Report

General thoughts The reviewed article is interesting, it points to interesting elements related to the study of the relationship between the participation of farmers' organizations and the certification of agricultural products. I would like to point out a few elements that should help the authors improve the study. In the opinion of the reviewer, the structure presented in the article requires the abstract to be organized and the discussion separated as a separate part, which should ensure its greater transparency. Check text and literature formatting. Abstract Abstract (needs to be structured) should include 1) research problem (the reviewer does not see it in the abstract), 2) the purpose of the work (it should be indicated more clearly), 3) research method and area (mention synthetically), 4) general description of the research results (general conclusions without describing them, what is their contribution to business practice or science) Admission Generally, the authors lead the reader into the analyzed research problem. In the introduction, you can find the purpose of the study, the duration of the study, an indication of the choice of the research method, and the subject / subject of the study. Lack of research questions or research hypothesis makes it difficult to evaluate the article. review of the literature The reviewer assumes that this part consists of chapters 2 and 3 Quite an interesting part of the study, it could be extended to a greater extent with European literature (if it is possible to indicate it). This part could be supplemented with a reference to lagging regions, intra-regional differentiation, of course in the aspect of the problem undertaken in the analysis. Supplement the literature review (the appropriate occurrence of the knowledge gap in the topic under study was not emphasized) Method and material Why did the authors choose the indicated methods of analysis and what are their disadvantages and advantages. Why were the indicated variables selected for the study (the need to explain their choice, their impact on the research topic). The results It seems to be a better solution to split the results and discussion into two separate parts It seems necessary to change the title of point 5.1. Discussion (not separated in the article separately) Presenting the discussion seems to be a consequence of the introduction and review of the literature and the research carried out. Can the results of the studies obtained by the authors be more closely related to other studies by the authors? Can the obtained results be compared with analyzes from other countries? Can the obtained research results be used in activities in other countries? Conclusions In addition to general conclusions, references to the results obtained, an indication of the problems that the authors had during the analysis, or who could use the research, In the part concerning conclusions, please indicate the originality of the presented research. Do the results of the analysis provide progress in the current knowledge and to what extent? Can the results of the analysis be used in other countries and to what extent? Can the research be related to international literature and to what extent? 

Reviewer 3 Report

The paper has made an effort to find out determinants of farmer's participation in three different types of certification against no certification employing a multinomial logistic regression model. A Farmer's membership in two types of farmer's association is one of the several explanatory variables. While going through the paper, I have two fundamental questions

  1. Why should farmers participate in certification? unless they anticipate the higher profit through premium price or they fear losing market because of food safety concerns on the part of consumers or policymakers. How have been these two factors in the case of Taiwan is not clear. No realized profit gain or no fear of losing marking even without certification would have been the reason for the low rate of participation (merely 32% in total or just 4% if you see the organic certification) in certification, despite the certification initiated long back i.e., 1994.
  2. I sense a high possibility of endogeneity in the empirical model, hence suggest going for some other alternative empirical tool or approach such as IV or selection model.

Please refer to the pdf file for other specific comments.

Round 2

Reviewer 3 Report

Thank you so much for addressing all the comments. I am quite convinced with the response to all the comments.